# Effect of Suppressive Levothyroxine Therapy on Bone Mineral Density in Young Patients with Differentiated Thyroid Carcinoma

**DOI:** 10.3390/metabo12090842

**Published:** 2022-09-07

**Authors:** André Borsatto Zanella, Laura Marmitt, Tayane Muniz Fighera, Rafael Selbach Scheffel, Poli Mara Spritzer, José Miguel Dora, Ana Luiza Maia

**Affiliations:** 1Thyroid Unit, Endocrinology Division, Hospital de Clínicas de Porto Alegre, Medical School, Universidade Federal do Rio Grande do Sul, Rua Ramiro Barcelos, 2350, Porto Alegre 90410-000, Brazil; 2Gynecological Endocrinology Unit, Hospital de Clínicas de Porto Alegre, Medical School, Universidade Federal do Rio Grande do Sul, Porto Alegre 90410-000, Brazil; 3Department of Pharmacology, Instituto de Ciências Básicas da Saúde, Universidade Federal do Rio Grande do Sul, Porto Alegre 90050-170, Brazil; 4Department of Physiology, Instituto de Ciências Básicas da Saúde, Universidade Federal do Rio Grande do Sul, Porto Alegre 90050-170, Brazil

**Keywords:** differentiated thyroid carcinoma, pediatric patients, suppressive levothyroxine, bone density

## Abstract

Suppressive levothyroxine therapy (sT4) is a cornerstone in the management of differentiated thyroid cancer (DTC). Long-term sT4 may affect bone mineral density (BMD). We evaluated the effect of sT4 on the bone mass of young DTC patients. In this cross-sectional study, BMD was evaluated via dual-energy X-ray absorptiometry in DTC patients younger than 25 years at diagnosis and undergoing sT4 for ≥1 year. The two control groups comprised patients matched for sex, age, and body-mass-index who were thyroidectomized for indications other than DTC and undergoing L-T4-replacement therapy, and healthy individuals with no prior known thyroid disease. Ninety-three participants were included (thirty-one in each group). There were no differences in the mean age, female sex (77.4% in all groups), or BMI between the sT4 group and each control group. The median TSH level was lower (0.4 [0.04–6.5] vs. 2.7 [0.8–8.5] mIU/mL, *p* = 0.01) and the mean L-T4 mcg/Kg levels were higher (2.4 ± 0.6 vs. 1.6 ± 0.3, *p* = 0.01) in the sT4 group compared to the L-T4-replacement therapy group. Lumbar spine, femoral neck, and total femur BMD were all similar among the groups. sT4 does not impact BMD in young DTC patients after a median time of suppression of 8 years. These findings may help in the decision-making and risk/benefit evaluation of sT4 for this population.

## 1. Introduction

Differentiated thyroid cancer (DTC) is the most common malignancy of the endocrine system [1]. Incidence is increasing at a rate of approximately 2% per year, particularly in the age group of 10 to 18 years [2]. According to the American Thyroid Association (ATA) guidelines, the initial management of pediatric DTC patients consists of total thyroidectomy, radioactive iodine (RAI), and suppressive levothyroxine therapy (sT4) [3].

The long-term suppression of thyrotropin-stimulating hormone (TSH) with supraphysiological doses of levothyroxine (L-T4) is a cornerstone of the treatment of DTC, since it reduces the risk of TSH-induced tumor growth and proliferation [3]. The clinical objective of this therapy is to decrease the risk of thyroid cancer recurrence and even cancer-related mortality, although the data supporting this approach are derived mainly from observational studies. In 1977, Mazzaferri et al. [4] retrospectively evaluated data from a multicenter study and demonstrated that, in 5 years, recurrence rates were significantly lower for patients with TSH suppression therapy compared to patients with no adjuvant therapy (10% vs. 40%, respectively, *p* < 0.01). Other studies have reported similar results [5,6,7]. A cohort study that included 867 patients with intermediate- and high-risk DTC showed no benefit of TSH suppression on progression-free survival or overall survival [8]. The only randomized clinical trial that evaluated whether TSH suppression decreases recurrence in DTC patients was performed in Japan in 2010 [9]. Patients were randomly assigned to receive L-T4 to suppress TSH (<0.01 mUI/L) or to keep TSH in the normal range. Disease-free survival was not inferior by more than 10% in the comparison of both groups, with more than 400 patients included. There are no data comparing risks and benefits in children for different degrees of TSH suppression [3].

Since thyroid hormone directly affects bone cells by stimulating the activity of osteoclasts and osteoblasts with a predominance of bone resorption and decreased bone mineral density (BMD) [10], the long-term administration of supraphysiological doses of L-T4 may cause serious side effects, including osteoporosis [11]. Bone turnover markers are significantly higher in long term suppressive levothyroxine therapy (sT4) patients, suggesting greater bone reabsorption in this group compared to controls [12,13]. Histomorphometry studies have shown that, in the presence of excess thyroid hormone, structural changes are observed in bone tissue with a reduction in trabecular volume, as well as increased porosity and reduced thickness of the cortical bone [14]. Similarly, a recent narrative review concluded that, in postmenopausal women, sT4 was associated with bone loss, the deterioration of bone architecture, and, ultimately, increased risk of fractures [15]. Additionally, a previous study of Wang et al. [16], which included 771 DTC patients with low and intermediate risk of recurrence, demonstrated that postoperative osteoporosis was increased among women with suppressed TSH compared to those in whom it was not suppressed (HR: 3.5, *p* = 0.023, 95%CI: 1.2–10.2).

Remarkably, most studies on the effects of thyroid hormones on bone have focused on adult patients. In children and young individuals, puberty is an important time for bone acquisition toward the attainment of peak bone mass by the end of skeletal development in the third decade of life [17,18]. Optimal peak bone mass acquisition is a determinant of osteoporosis risk and osteoporotic fracture risk [19]. Although bone strength is largely determined by genetics, environmental factors such as dietary intake, physical activity, and drugs also affect the acquisition of bone mass and the achievement of individual genetic potential [20]. Thus, using sT4 in this period could have a negative effect on bone metabolism and an impact on bone health during life.

In this study, we aimed to evaluate the effect of long term sT4 on bone mass in a population of children, adolescents, and young adults with DTC, followed in a tertiary, university-based referral center. The clinical and laboratory parameters associated with sT4 and their correlations with BMD were evaluated.

## 2. Results

### 2.1. Patients

Ninety-three participants, with thirty-one in each group, were evaluated. Their clinical and oncological characteristics are described in Table 1. The sT4, L-T4 replacement, and healthy groups had similar ages (28.3 (range of 18–38) vs. 29.1 (17–42) vs. 28.6 (20–39), *p* = 0.86)), sex (female 77.4% vs. 77.4% vs. 77.4%, *p* = 1.00), and body mass index (BMI) (25.5 ± 5.1 vs. 25.6 ± 4.3 vs. 24.1 ± 3.6 kg/m^2^, *p* = 0.33). The mean age of LT4 start in the sT4 group was 19.1 (range 6–25) years and 24.0 (5–41) years in the L-T4-replacement group (*p* = 0.01). The median time of suppression was 8.0 (3.0–12.0) years in the sT4 group, and the median time of levothyroxine use in the L-T4-replacement group was 2.0 (1.0–8.0) years, *p* = 0.014.

The mean LT4 dose was higher in the sT4 group than in the L-T4-replacement control group (2.4 ± 0.6 vs. 1.6 ± 0.3 mcg/kg, *p* = 0.01). Accordingly, the median TSH level was lower in the sT4 group (0.4 [0.04–6.5] vs. 2.7 [0.8–8.5] mIU/mL, *p* = 0.01). However, no statistically significant difference was observed for the healthy individual group (n = 13), TSH = 1.77 (1.14–3.08 mIU/L). Of the 31 patients included in the sT4 group, 30 (96.8%) had papillary thyroid cancer (PTC) and 1 (3.2%) had follicular thyroid cancer (FTC). According to the TNM-8 and the ATA risk classification systems, the patients in the sT4 group were classified as follows: 24 patients on stage I and 7 on stage II; 10 patients as low risk, 14 as intermediate risk, and 7 as high risk, respectively.

In the L-T4-replacement group, 13 (42.0%) patients had medullary thyroid cancer (MTC), 10 (32.2%) had benign thyroid disease, 7 (22.6%) had PTC, and 1 (3.2%) had FTC. The rates of post-surgery hypoparathyroidism (25.8% vs. 16.1%, *p* = 0.5), smoking (3.2% vs. 3.2%, *p* = 1.0), and oral contraceptive use (33.3% vs. 41.7%, *p* = 0.7) were similar in the sT4 and L-T4-replacement groups, respectively. In the healthy control group, smoking was reported by only one participant (3.2%). There were no differences in the serum levels of calcium corrected by albumin (8.5 ± 0 + 5 vs. 8.8 ± 0.5, *p* = 0.1), phosphorus (3.6 ± 0.6 vs. 3.8 ± 0.5, *p* = 0.4), or parathyroid hormone (PTH) (29.9 ± 9.3 vs. 40.1 ± 14.9, *p* = 0.1) between the sT4 and L-T4-replacement groups, respectively.

### 2.2. Bone Mineral Density

There were no differences in the BMD values and Z-score in lumbar spine in the sT4 group when compared with the L-T4-replacement or healthy control groups. Similar results were found for femoral neck and total femur (Table 1 and Figure 1). Six patients (three in the sT4 and three in the L-T4-replacement group) who underwent total body less head analysis also presented similar results in the sT4 group vs. the control group regarding BMD and Z-score (1.004 ± 0.561 vs. 1.008 ± 0.416 g/cm^2^, *p* = 0.93 and 0.1 [−1.5–not available] vs. 0.1 [−0.6–not available], *p* = 1.0). There were two patients (6.5%) with low bone mass in the L-T4-replacement group and none in sT4 group (*p* = 0.49). Similar results were found when the sample was stratified with lower cutoff points for age of onset of suppression (data not shown).

We also performed a correlation between the time of suppression and a possible association with lower BMD values. The results found no significant differences in the BMD values, even in patients with longer time on sT4. Bone mass in lumbar spine, neck, and total femur for the sT4 group vs. the L-T4-replacement group were similar (r = 0.04, *p* = 0.8 and r = −0.02, *p* = 0.8; r = −0.06, *p* = 0.7 and r = −0.11, *p* = 0.5; r = −020, *p* = 0.2 and r = −0.02, *p* = 0.9, respectively). Further correlation analysis between the median TSH and free T4 levels with the z score of the femoral neck failed to demonstrate an association (r = 0.1, *p* = 0.59 and r = −0.2, *p* = 0.39, respectively).

### 2.3. Fractures

Six patients in the sT4 group (19.3%) and seven patients in the L-T4-replacement group (22.3%) reported past fractures. However, only one patient in each group had a fracture after a thyroidectomy. There was one elbow fracture in the sT4 group and one wrist fracture in the L-T4-replacement group, both secondary to trauma and considered as nonpathological fractures. We were unable to obtain information on when and how fractures occurred in three patients in the sT4 group and four patients in the L-T4-replacement group. In the healthy control group, no fragility fractures were reported.

## 3. Discussion

Suppressive levothyroxine therapy has been advocated for as part of treatment in pediatric DTC patients. In this study, we showed that children, adolescents, and young adults who underwent sT4 had no decrease in BMD when compared to thyroidectomized patients receiving replacement doses of L-T4 and healthy individuals with no history of thyroid illness, matched for sex, age, and BMI.

Thyroid hormones have a fundamental role in the regulation of bone turnover, and many studies have evaluated the effects of their excess on bone remodeling and on the risk of osteoporosis and fragility fractures in young adults [14,21,22]. The effects of L-T4 on bone metabolism in children have been investigated, with conflicting results [23,24]. Demartini et al. [23] demonstrated that total body BMD was significantly decreased in children aged 7 to 14 years with congenital hypothyroidism receiving L-T4 treatment. However, a 2009 study conducted by Salama et al. [24] including 35 children undergoing treatment with L-T4, with a mean age of 11.5 years, found no differences in the Z-score and BMD, nor any changes in the bone markers. It is important to note that these patients used substitutive doses of L-T4, with a mean TSH level of 4.48 mIU/L, and BMD assessment included only lumbar spine and proximal femur. A population-based prospective cohort study evaluated the association of thyroid function with bone mass during childhood [19]. BMD and thyroid function were assessed in 4204 children aged 6 to 10 years, without previous thyroid disease. After controlling for multiple covariables, the study showed that higher free thyroxine was associated with lower bone mass at the ages of 6 and 10 years.

Interestingly, few data are available on sT4 in children and adolescents, and the impact of supraphysiological doses of L-T4 on peak bone mass in young patients has been poorly studied. A 2015 study evaluated whether sT4 in a pediatric population had a negative effect on volumetric BMD and bone microarchitecture, assessed using high-resolution peripheral quantitative computed tomography [25]. Seventeen DTC patients undergoing sT4 since adolescence were compared to 34 healthy volunteers, matched for age, sex, and BMI. The mean age of L-T4 start was 12.6 years, with a mean duration of treatment of 12.4 years. No differences were found between the groups with respect to BMD and Z-score. These data suggest that long-standing sT4 during peak bone mass had no significant adverse effect on bone density. Our data showed similar results regarding the BMD and Z-score, in accordance with those previous observations. Additionally, our study found no evidence of a clinical effect of sT4, since we observed no fragility fractures in the sT4, L-T4-replacement and healthy control groups, although this information was lacking for seven participants.

There are many known factors that can interfere with peak bone mineral accretion, which occurs around the age of 12 years in girls and 14 years in boys, including heredity, fractures during childhood, nutrition, exercise, oral contraceptive, and age at menarche [18,26]. During the pubertal period, estradiol has important effects, not only for increasing BMD but also for attenuating bone turnover at the endocortical surface, leading to an increase in cortical thickness [27]. The available evidence suggests that sT4 is associated with bone loss in postmenopausal women but not in premenopausal women and in men. In this context, sT4 safety can be explained by the protective effect of sexual steroids on bone remodeling in young individuals [14]. The knowledge that sT4 starting before peak bone mass does not decrease bone density is reassuring for clinicians and patients who have been contemplating this therapy for long periods.

Our study has several methodological aspects that merit consideration. First, all the patients included in this analysis were attended a follow-up at the same institution, and BMD was measured using the same equipment and evaluated by the same operator. We are aware that data about bone microarchitecture and bone turnover markers would provide more robust results but, unfortunately, they were unavailable. Although 31 patients may not be considered a sizeable population, we understand that this sample size is compatible with the rarity of this disease and is in line with other studies published on this subject. Unfortunately, due to the limited number of patients, subgroup analysis according to ATA risk, age categories, and the degree of TSH suppression where not allowed. Of note, there were differences in the mean age of levothyroxine start in the sT4 and replacement group. On the other hand, the relatively small number of patients allowed for a detailed analysis of individual responses, which is usually not possible in more extensive studies, providing important information for clinical care. Of note, we did not perform a questionnaire about dietary calcium consumption and did not measure vitamin D levels. Physical activity, a potential confounder, was not quantitated in this study, but no participants had physical limitations or practiced professional sports. The high rate of hypoparathyroidism found in our sample (20%) and a possible effect of the use of calcium and calcitriol on the results were also highlighted. Additionally, data on calcium metabolism in the group of healthy individuals were not available. Lastly, time on suppressive therapy was considered the time between surgery and the last follow-up. Although we cannot guarantee that patients actually remained in suppression for the entire period, this ensures a more real-life scenario.

In conclusion, our study indicates that sT4 in children, adolescents, and young adults with DTC has no negative effect on bone mass. Moreover, when we performed this analysis considering the TSH suppression time, we found no differences in bone mass and Z-score in the lumbar spine, neck, and total femur. In clinical practice, these findings have an important impact, reassuring clinicians and patients with regard to sT4. Further long-term studies are needed, with a larger sample and longitudinal follow-up, to determine the effect of sT4 in patients of different ages, especially younger ages.

## 4. Material and Methods

### 4.1. Patients and Study Design

DTC patients treated at the Thyroid Outpatient Clinic of the Thyroid Unit at Hospital de Clínicas de Porto Alegre (HCPA), a tertiary care, university-based hospital in southern Brazil, were recruited for this cross-sectional study. From 2015 to 2018, consecutive patients with a histological diagnosis of DTC under the age of 25 years, who had undergone suppressive LT4 therapy for at least 1 year, were included in this study (sT4 group). Thyroidectomized patients undergoing L-T4-replacement therapy and healthy individuals (no previous history of thyroid disease) matched for sex, age, and BMI were used as group controls. A matched control group was established by matching each patient with a control of the same gender, and of similar age and BMI. Age and BMI were matched as continuous variables. Regarding age, patients were divided into three groups: children (≤12 years old), adolescents (12 to 18 years old), and adults (>18 years old). The patients with DTC included in the L-T4-replacement group underwent BMD during the first year after surgery and also had no suppressed TSH (<0.1 mIU/mL) between the thyroidectomy and the exam. The study exclusion criteria were hyperparathyroidism, bone metastasis, and history or current use of medications that affect bone mass, such as glucocorticoids and bisphosphonate. In the healthy control group, oral contraceptive use was an exclusion criterion.

### 4.2. Treatment Protocol and Follow-Up

Our DTC treatment protocol consists of total thyroidectomy, followed by administration of RAI as indicated and sT4 according to the current guidelines [3,28,29]. The ATA pediatric guideline recommends that the initial TSH goal should be tied to ATA pediatric risk level and current disease status: low risk (0.5–1.0 mIU/L), intermediate risk (0.1–0.5 mIU/L), and high risk (<0.1 mIU/L) [3]. In children with evidence of an excellent response, TSH can be normalized to the low–normal range [3].

### 4.3. Outcomes

The main outcome was BMD, measured using dual-energy X-ray absorptiometry (DXA), with a Z-score ≤ −2.0 indicative of low bone mass [30]. The findings observed in the sT4 group were compared with those obtained in L-T4-replacement or healthy control groups. Time on sT4 was considered that between thyroidectomy and the last follow-up while on suppressive therapy according to individual response to therapy (TSH ≤ 0.1 mIU/mL). Patients without suppressed TSH in the follow-up consultations had their L-T4 dose adjusted, but the time of suppression was still counted for “intention-to-treat” purposes. We also evaluated the number of fractures between the three groups. We consider a fracture non-pathological when it occurs in a bone with no known disease, which may or may not be related to trauma.

### 4.4. Clinical and Oncological Features

All the data were collected directly through patient interviews and medical record review. The demographic data evaluated were sex, age, skin color, weight, height, BMI, history of previous fractures, smoking, alcohol, and medication use. In addition, the serum total calcium, albumin, phosphorus, and PTH measurements were annotated. Tumor characteristics such as histological type, size, presence of lymph node, or distant metastases, and follow-up time, were also collected. The staging systems used were the 8th edition of the American Joint Committee on Cancer: Tumor, Node, Metastasis (AJCC/TNM-8) and the ATA risk stratification [3,31,32].

### 4.5. Laboratory Analysis

Serum PTH was evaluated using the chemiluminescent microparticle immunoassay method through the ARCHITECT ci 4100 equipment (Abbott Diagnostics, Abbott Park, IL, USA), with reference values (RVs) of 15.0–68.3 pg/mL; the intra-assay and inter-assay coefficients of variation were 6.1 and 6.4%, respectively. Total serum calcium was evaluated using the NM-BAPTA method and corrected via albumin levels (corrected calcium = 0.8 × (4.0−serum albumin) + serum calcium) with RVs of 8.6 to 10.0 mg/dL. Serum phosphorus was evaluated using the Molybdate UV and colorimetric (xylidil blue) methods, with RVs of 2.5–4.5 mg/dL. The electrolyte tests were conducted using Cobas 8000 c702 equipment (Roche Diagnostics, Indianapolis, IN, USA). Free thyroxine and TSH were evaluated using a chemiluminescence immunoassay (Abbot, Chicago, IL, USA), with RVs of 0.7–1.48 ng/mL and 0.35–4.94 mUI/L, respectively. These tests were all conducted in the HCPA central laboratory.

### 4.6. Bone Mineral Density

All the patients underwent BMD evaluation through DXA using a Lunar Prodigy Primo device (Encore version 14.10, Radiation Corporation, Madison, WI, USA), which was analyzed by the same operator (TMF). The sites analyzed were lumbar spine (L1–L4), femoral neck, total femur, and total body less head (for patients under 18 years of age). Bone mineral density was expressed in g/cm^2^ and Z-score. The Z-scores for BMD were calculated with age-matched controls from the third National Health and Nutrition Examination Survey (NHANES III). The coefficient of variation for lumbar spine and femur was 0.022 g/cm^2^ and 0.033 g/cm^2^, respectively. For this calculation, we measured 30 patients two times, repositioning the patient after each scan.

### 4.7. Statistical Analysis

Continuous variables were expressed as means ± standard deviation (SD) for variables with normal distribution, and medians and interquartile ranges (P25–75) for variables with non-Gaussian distribution. Categorical variables were expressed as frequency and percentage. Comparative analyses of frequencies were performed using the Pearson chi-square test, ANOVA, or Kruskal–Wallis as appropriate. Pearson correlation was used to evaluate a possible association between the time of suppression in years as a continuous variable and the BMD results. These analyses were performed using the Statistical Package for the Social Sciences Professional software, version 20.0 (IBM Corp., Armonk, NY, USA). All the tests were two-tailed, and *p* < 0.05 was considered statistically significant.

## Figures and Tables

**Figure 1 metabolites-12-00842-f001:**
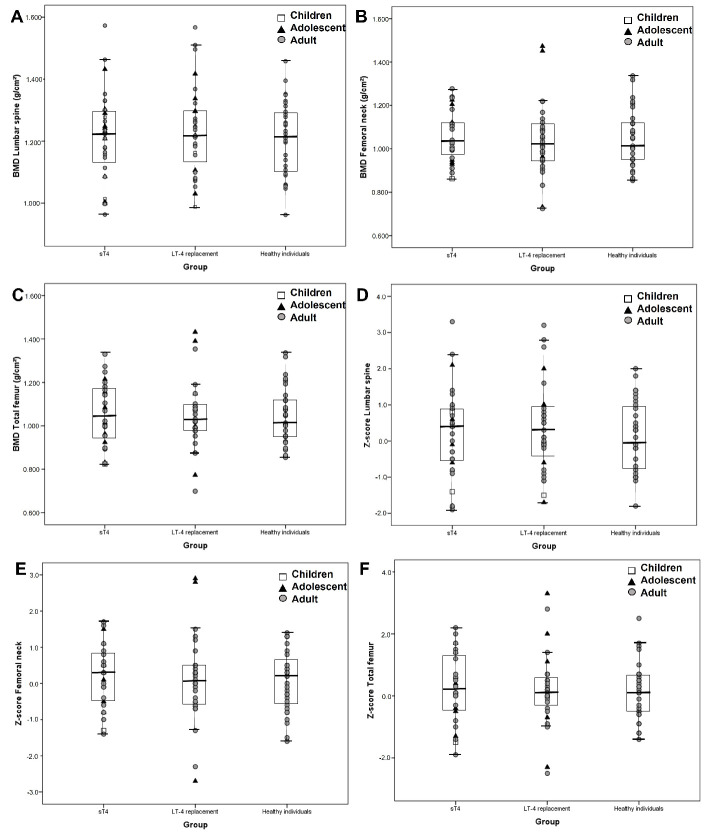
Bone mineral density and Z-score data with no significant differences between healthy control and the sT4 and L-T4-replacement groups. (**A**): BMD lumbar spine; (**B**): BMD femoral neck; (**C**): BMD Total femur; (**D**): Z-score lumbar spine; (**E**): Z-score femoral neck; (**F**): Z-score total femur. Data stratified by age * in three groups (children, adolescents, and adults). BMD: bone mineral density; sT4: suppressive levothyroxine therapy; L-T4: levothyroxine. * In sT4 and L-T4-replacement groups, age was considered as patients started to use the medication.

**Table 1 metabolites-12-00842-t001:** Clinical and oncological characteristics of 93 participants included in this study.

Characteristics	s-T4 (n = 31)	L-T4 Replacement (n = 31)	Healthy Individuals (n = 31)	*p*
Age (years)	28.3 (18–38)	29.1 (17–42)	28.6 (20–39)	0.86
Female sex—n (%)	24 (77.4)	24 (77.4)	24 (77.4)	1.0
BMI (kg/m^2^)	25.5 ± 5.1	25.6 ± 4.3	24.1 ± 3.6	0.33
Age at diagnosis (years)	19.1 (6–25)	24.0 (5–41)		0.01
Histological type				
Papillary—n (%)	30 (96.8)	7 (22.6)		
Follicular—n (%)	1 (3.2)	1 (3.2)		
Medullary—n (%)	0 (0.0)	13 (42.0)		
Benign—n (%)	0 (0.0)	10 (32.2)		
Tumor size (cm)	2.0 (1.2–2.7)	2.5 (2.1–3.5)		
Distant metastasis	7 (22.5)			
ATA risk				
Low—n (%)	10 (32.3)			
Intermediate—n (%)	14 (45.2)			
High—n (%)	7 (22.5)			
TSH (mIU/mL)	0.4 (0.04–6.5) †	2.7 (0.8–8.5) †	1.77 (1.14–3.08)	† 0.01
Levothyroxine (mcg/kg)	2.4 ± 0.6	1.6 ± 0.3		0.01
Hypoparathyroidism—n (%)	8 (25.8)	5 (16.1)		0.5
Lumbar spine BMD (g/cm^2^)	1.212 ± 0.142	1.231 ± 0.141	1.205 ± 0.120	0.74
Z-score	0.4 (−0.6–0.9)	0.3 (−0.6–1.0)	0.3 (−0.8–1.0)	0.74
Femoral neck BMD (g/cm^2^)	1.059 ± 0.115	1.040 ± 0.170	1.041 ± 0.121	0.85
Z-score	0.3 (−0.5–0.8)	0.0 (−0.5–0.5)	0.2 (−0.6–0.8)	0.84
Total femur BMD (g/cm^2^)	1.055 ± 0.144	1.046 ± 0.162	1.041 ± 0.131	0.93
Z-score	0.2 (−1.4–1.3)	0.1 (−0.3–0.6)	0.1 (−0.5–0.7)	0.93

Data are expressed as mean ± standard deviation or range, median (25–75th percentiles), or number (frequencies). sT4: suppressive levothyroxine therapy; L-T4: levothyroxine; BMI: body mass index; ATA: American Thyroid Association; TSH: thyrotropin-stimulating hormone; BMD: bone mineral density. † statistically different.

## Data Availability

Not applicable.

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
