# Peer review of "Effect of Suppressive Levothyroxine Therapy on Bone Mineral Density in Young Patients with Differentiated Thyroid Carcinoma"

_metabolites, 2022, doi:10.3390/metabo12090842_

Round 1

Reviewer 1 Report

This paper does not add new pieces of relevant information to the current knowledge on the issue of the effect of suppressive levothyroxine therapy on bone mineral density in young patients with thyroid carcinoma. The study describes findings on a limited sample albeit in line with the numbers in the literature on this field. Moreover, statistical methods are inadequate and results are poorly reported.

Specific point:

-The impact of age and treatment dosage are not explored and the association with tratment duration/treatment cumulative dose is not supported by numbers.

-Details on the matching process used for age and BMI should be added. E.g. is age matched in continuous or in classes?.

-The tests used for comparison do not account for the nature of the study design that is matched.

-Figure legend should be enriched to describe the features of the box plot. Means should be indicated. Individual data points might be reported differentiated in the 3 age classes (i.e. children, adolescent and adults) to support the conclusions (page 5 line 51).

-Less emphasis should be put in the discussion on the obtained results since with data on 30 subjects per group there is no sufficient evidence to "demostrate", but only to suggest.

Reviewer 2 Report

In the manuscript entitled “Effect of suppressive levothyroxine therapy on bone mineral  density in young patients with thyroid carcinoma”, considering the role of thyroid hormones in the regulation of bone turnover, Zanella AB et al. evaluated the effect of farmacological long-term suppression of TSH on bone mass, in a population of young DTC patients including children, adolescents, and young adults.

They demonstrated that the therapy had not negative effects on bone mass, don’t finding difference with groups of patients thyroidectomized or healthy individuals with no history of thyroid illness

The paper is well written

In the introduction section the clinical problem is well focused, and the relevant literature available is well presented in the discussion

The study is well conducted and presented

Reviewer 3 Report

Title - Effect of suppressive levothyroxine therapy on bone mineral density in young patients with thyroid carcinoma

General Comments

The aim of the manuscript was to evaluate the effect of suppressive levothyroxine therapy on bone mineral density in young patients with thyroid carcinoma, in a cross-sectional study.

The article has structural problems, being unclear regarding the groups studied, the comparisons between them, and the reason for their inclusion; this makes it difficult to read and understand. Also, it needs English editing, which I will not address in this review. In this form, it is not suitable for publication.

Specific Comments

Abstract

1. Lines 24-25 – “Patients… groups.” This sentence is difficult to understand. I only realized what the authors meant after reading the entire article.

2. Line 26 – The sentence that follows “Ninety-three patients were included (31 in each group).” is also unclear, because the healthy controls are not patients.

3. Line 28 – “…between groups.” Which groups? Since the authors present two p-values for each variable, they have to clarify which group comparison they are making.

4. Line 30 – “…sT4 patients.” At this point, it is not clear which group the authors are referring to.

Introduction

5. Line 40 - “…childhood…” This is also true for adulthood, right? This sentence is misleading.

6. Lines 54-55 – “The only randomized clinical trial that evaluated if TSH suppression decreases recurrence…” I do not agree with the authors. For instance, see this paper, published in 2021, “A Multicenter, Randomized, Controlled Trial for Assessing the Usefulness of Suppressing Thyroid Stimulating Hormone Target Levels after Thyroid Lobectomy in Low to Intermediate Risk Thyroid Cancer Patients (MASTER): A Study Protocol”.

Results

7. Line 84 - See previous comment 2.

8. Line 85 – “As expected…” This kind of consideration should be presented in the Discussion section, and this consideration should be explained.

9. Table 1 – Two p-values are presented in the P column, so the authors need to refer which comparison they are referring to.

10. Line 97 – “…than in those in L-T4 replacement controls…” Written this way gives the idea that there are two control groups taking levothyroxine.

11. Lines 98-99 – “Accordingly, the median TSH level was lower in the sT4 group (0.4 [0.04-6.5] vs. 2.7 [0.8-8.5] mIU/mL, P = 0.01” - The authors should explain why they have used “mean” or “median”. I have checked, and this information is also not given in the Material and Methods section.

12. Line 105 – “In the L-T4 replacement group, 13 (42.0%) patients had medullary thyroid cancer…” I do not understand this. If 13 of these patients also had cancer, why were they not included in the sT4 group? Furthermore, why is there no information in Table 1 about the size of the tumors of these patients? The reader gets the idea that these patients only had benign disease. Why are they a control group?

13. Lines 107-109 – “…smoking (3.2% vs. 3.2%, P = 1.0), and oral contraceptive use (33.3% vs. 41.7%, P = 0.7) were similar in the sT4 and L-T4 replacement groups…” And what about the healthy group?

14. Lines 110-112 – “There… respectively.” And what about the healthy group?

15. Lines 142-148 – Why there is no information on the frequency of fractures in the healthy group? It is very important to make this comparison, because this was the group that did not take levothyroxine.

Discussion

16. Lines 181-182 – “Also, our study found no evidence of clinical effect of sT4, since we observed similar rates of fracture in sT4 and L-T4 replacement groups.” But what about the healthy control group? The rates of fracture are similar but could be higher than in the control group, leading to a clinical effect of both sT4 and L-T4.

17. Lines 219-220 – “Also, our study found no evidence of clinical effect 181 of sT4, since we observed similar rates of fracture in sT4 and L-T4 replacement groups.” Considering my previous comments, I believe this sentence is an overstatement.

Material and Methods

18. Lines 227-228 – “Thyroidectomized patients undergoing L-T4 replacement therapy…” Information about the reasons for thyroidectomy are needed.

19. Lines 235-238 – “The … (3)” In the present paper, how many patients were in each group?

20. Lines 283-285 – When using medians and percentiles why did the authors not use nonparametric tests?

Reviewer 4 Report

Dear Authors,

This is an interesting study.

Here are my observations/suggestions/comments:

1, Title – please replace “thyroid carcinoma” with “differentiated thyroid carcinoma” since the other types of thyroid cancers are not associated with the indication of suppressive therapy  

2. Abstract - TSH was lower when compare to each of the control groups?

3. Abstract – I do not entirely agree with “sT4 does not impact BMD in young DTC patients.”  - I suggest you introduce here the mean period of time of suppressive therapy.

Actually you do not know the effect on peak bone mass since the patients are so young.

4. Abstract/conclusion – I suggest that studies with a lager cohort, a longitudinal approach, including patients of different ages, are the next logical step of your study

5. Introduction – Differentiated thyroid cancer is the most frequent endocrine malignancy regardless the age (including in young patients as seen in your study)

6. Introduction/discussion – you should also comment the effect of long term suppressive therapy on terms of bone turnover markers

7. Results – Do you have any data on lumbar DXA – based trabecular bone score?

8. Methods – there is not Method section

9. Results - Did the entire cohort have the central DXA assessment done on the same device? Did you have the value of least significant change?

10. Did you take an ethical precaution on performing DXA in control groups (young individuals who are not typically candidates to DXA, a procedure  which is based on X ray exposure).

Best regards,

Thank you,

Round 2

Reviewer 1 Report

I still maintain my concerns about the relevance of the paper. Moreover, the authors have only partially answered to my comments.

1)     Paired tests that account for matching should have been used

2)     No changes were done on Figure 1 that is without mean and individual data point (differentiated by age group). The legend is lacking of an explanation of the details in the box-plot.

Reviewer 3 Report

R#2 - Effect of suppressive levothyroxine therapy on bone mineral density in young patients with thyroid carcinoma

General Comments

The manuscript has improved. Still, there remain several points which need to be addressed.

Specific Comments

Abstract

1. Line 30 – Please replace “both control” with “each control groups”, since comparisons were made with one control group at a time, as they have two different p-values.

Introduction

2. In fact, the paper A Multicenter, Randomized, Controlled Trial for Assessing the Usefulness of Suppressing Thyroid Stimulating Hormone Target Levels after Thyroid Lobectomy in Low to Intermediate Risk Thyroid Cancer Patients (MASTER): A Study Protocol” does  not present results, but there are other papers, though not RCTs, reporting valuable information, such as this ones “Thyrotropin Suppression Increases the Risk of Osteoporosis Without Decreasing Recurrence in ATA Low- and Intermediate-Risk Patients with Differentiated Thyroid Carcinoma” and “Association of Thyrotropin Suppression With Survival Outcomes in Patients With Intermediate- and High-Risk Differentiated Thyroid Cancer”, which present a different point of view. These are just examples; I believe this issue should be better presented since it is an important point for the manuscript. Moreover, why did the authors not choose to do an RCT, since this information is lacking?

Results

3. Lines 95-99 - The mean age of LT4 start was significantly different between the sT4 and the LT-4 replacement group, as well as the median time of levothyroxine. Based on this significant difference, how can group LT-4 be a control group of the sT4 group regarding levothyroxine use?

4. Previous comment 12 – I still do not understand why 7 patients with PTC and one with FTC were included in the LT-4. What were the differences between these patients and the ones with PTC and FCT included in the sT4 group? Again, why there is no tumor size for the tumor cases included in the LT-4 group?

5. Lines 119-121 – If no oral contraceptive use was an exclusion criterion to be considered a healthy individual, this should be mentioned in the Material and Methods section, and not in the Results section.

6. Line 125 – “There were no differences on BMD values and Z-score in lumbar spine the sT4 when…” – There is something missing in this sentence.

7. Previous comment 15. I do no understand why the authors do not have information about traumatic and non-pathological fractures, and have information about fragility fractures (healthy group). Additionally, the distinction between non-pathological fracture and traumatic fracture is unclear.

8. Lines 161-162 – “only 1 patient in each group had a fracture after thyroidectomy” How many years after thyroidectomy? Was it a fragility fracture? The authors   were unable to obtain information on when and how fractures occurred in 3 patients in the sT4 group and 4 patients in the L-T4 replacement group, which makes the information about fractures not very useful.

Discussion

9. Lines 200-201 – “Also, our study found no evidence of clinical effect of sT4, since we observed similar rates of fragility fractures in sT4, and L-T4 replacement and healthy control groups” I think this sentence should be rewritten since, as far as I could understand, no fragility fractures were found.
